# Dependence of Structural and Optical Performance of Lanthanum Fluoride Antireflective Films on O Impurities

**Wenli Lu** [1,2], **Bo Pan** [2,3], **Ruiying Miao** [1,2], **Bingzheng Yang** [1,2], **Chuang Yu** [1,2,*], **Dongwei Zhang** [2,3], **Dehong Chen** [1,2], **Liguo Han** [1,2] and **Zhiqiang Wang** [1,2,3]

1   GRIREM Advanced Materials Co., Ltd., Beijing 100088, China
2   National Engineering Research Center for Rare Earth, GRINM Group Co., Ltd., Beijing 100088, China
3   GRIREM Hi-Tech Co., Ltd., Langfang 065201, China
*   Correspondence: yyuchuang@163.com; Tel.: +86-10-61590666

**Abstract:** Lanthanum fluoride (LaF$_3$) thin films were deposited on the Ge substrate using the molybdenum boat evaporation method. The effect of films' oxygen impurity on the infrared optical properties has been investigated for the first time in this report. With the increase in oxygen content in the films, the F content decreases, and the O/F ratio decreases from 0.160 to 0.055. XRD patterns reveal that the presence of O impurity destroys the crystal structure integrity of the LaF$_3$ films and leads to the intensification of infrared absorption. The average transmittance decreases from 58.1% to 52.2%, and the peak transmittance decreases from 59.9% to 54.5%. Additionally, the refractive index and extinction coefficient of LaF$_3$ films with different oxygen content are obtained by fitting the transmittance test data. The results show that the refractive index and extinction coefficient of the films in 8–12 μm increase with the increase in oxygen content, the average refractive index increases from 1.339 to 1.478, and the extinction coefficient increases from 0.001 to 0.030. In this paper, the influence of oxygen impurity in the LaF$_3$ film on its infrared optical properties is revealed, which lays a theoretical foundation for the development of high-performance LaF$_3$ infrared antireflective film.

**Keywords:** oxygen impurity; LaF$_3$; infrared antireflective film

## 1. Introduction

Infrared materials and imaging are widely used in military and civil fields, such as night vision reconnaissance and guidance, infrared search and tracking, aerospace detection, road transportation, shipping and marine transportation, disaster prevention and mitigation, and many other aspects [1–5]. It is urgent to develop high-performance and low refractive index infrared film materials to reduce reflection loss of device surface and improve the transmittance of optical elements [6–13]. Lanthanum fluoride (LaF$_3$) material has excellent optical properties and good chemical stability [14,15]. It was widely used in the fluoride-glass ceramic [16,17], the scintillation crystals and rare earth crystal laser materials required by high-energy physics, nuclear medical imaging and industrial CT [18–20], arc lamp carbon electrode for lighting and other photoelectric fields, etc. [21–23]. Moreover, it has a low refractive index (1.4–1.6, compared with the Ge substrate) and low price. Hence, it has attracted much attention in the infrared materials field.

In previous studies, the LaF$_3$ material was often used as a high refractive index material (compared to the substrate material) and protective film for the ultraviolet and visible light regions. It exhibited excellent mechanical properties and has no moisture absorption [4,24–26]. Based on the above advantages and low refractive index of LaF$_3$, it was regarded as a promising antireflection coating material. In the early days, the mixed LaF$_3$-BaF$_2$ thin film was reported by Targove and Murphy [27] and the mixture film showed the fluorite crystalline structure of BaF$_2$. The large tensile stress of the pure LaF$_3$ film was greatly reduced and the infrared optical performance was improved by the addition of BaF$_2$.

In recent years, the LaF$_3$ film was admixed with a different amount of BaClF in Li's work [2]. Similarly, the internal stress of the BaClF-LaF$_3$ film was lower than the pure LaF$_3$ film and the excellent antireflection approach of the Ge surface was obtained by the mixed layer. Subsequently, Cheng's team investigated the evaporation properties of the LaF$_3$ material and its optical constants in 2.5–12 μm [14,15]. Then they used LaF$_3$ as a low refractive index material to prepare a high durability antireflective film in 3.7–4.8 μm on the Ge substrate. In summary, the previous studies mainly focused on reducing the internal stress and the refractive index of the LaF$_3$ film. But the relationship has rarely been explored between the impurities in the LaF$_3$ film and the far-infrared (8–12 μm) optical performance. Especially, the influence of oxygen impurities on the infrared optical performance of LaF$_3$ film is not clear. The cognitive gap in this aspect will limit the development of high-quality LaF$_3$ infrared antireflective coating materials.

In this paper, a single-layer LaF$_3$ film was deposited on the Ge substrate by means of the molybdenum boat evaporation method. By selecting the LaF$_3$ crystals with different oxygen content as raw materials and keeping other evaporation processes consistent, the LaF$_3$ monolayer infrared antireflective films with different oxygen content were obtained. The infrared optical properties of the film were analyzed by WQF-510A Fourier transform infrared spectrometer and TFCalc software. The surface chemical composition of the monolayer LaF$_3$ film was analyzed by the X-ray fluorescence spectrometer (XRF). Based on the above analysis results, the relationship between the oxygen impurity content in the film material and its long-wave infrared optical properties was obtained in this paper, which laid the foundation for the subsequent development of high-performance LaF$_3$ infrared antireflection coatings.

## 2. Materials and Methods

### 2.1. Raw Materials

LaF$_3$ with a purity of 99.99 wt% (oxygen impurity content is not calculated) is produced by GRIREM Advanced Materials Co., Ltd., and it is used as the evaporation source material in this work. The specific values of oxygen content in the LaF$_3$ raw materials are shown in Table S1.

### 2.2. Film Preparation

The LaF$_3$ films with different oxygen content were deposited on the Ge (111) substrate with a 25mm diameter and a 1mm thickness. They were prepared by the HF-1100 vacuum coating equipment (Chengdu Hangfan Vacuum Technology Co., Ltd., Chengdu, China). After cleaning agent, deionized water and ultrasonic pretreatment, the Ge substrate was put into the vacuum chamber. The pressure was below $2 \times 10^{-3}$ Pa. Additionally, in order to clean and activate the substrate surface, the preheated substrate was subjected to ion bombardment for 10 min before coating. In the process of film deposition, the ion-assisted technology was used to make the deposited molecules or atoms on the surface of the substrate obtain greater kinetic energy after the bombardment. The film material particles with sufficient kinetic energy had a high mobility, which increases the aggregation density of the film layer, thus improving the performance of the film layer. Ion source parameters were as follows: the plate voltage was 300 V, the electron beam was 60 mA and the ion beam was 40 mA. The Ar flow was 10 sccm, and the deposition temperature was 200 °C. LaF$_3$ was evaporated from the molybdenum boat, the evaporation current was 230 A and the evaporation rate was 3.5 nm /s. According to the different oxygen content, the obtained film materials are divided into five types, named Ge/LaF$_3$-1(13.33 at.%), Ge/LaF$_3$-2(9.60 at.%), Ge/LaF$_3$-3(8.99 at.%), Ge/LaF$_3$-4(8.30 at.%) and Ge/LaF$_3$-5(4.84 at.%).

### 2.3. Analysis Device

In order to obtain the optical properties of the LaF$_3$ films with the different oxygen content in the far-infrared (8–12 μm) region, the WQF-510A Fourier transform infrared spectrometer (Beijing hongzuo Shengwei Technology Co., Ltd., Beijing, China) was used.

The X-ray fluorescence spectrometer (ZSX Primus ii, Rigaku, Tokyo, Japan) was used to analyze the chemical composition of the films' surface. The X-ray powder diffractometer (CoKa1, λ = 1.78897, Rigaku, Tokyo, Japan) was used to characterize the crystallography structure of the films. Scanning electron microscopy (SEM, JSM-7610, Rigaku, Tokyo, Japan) was used to characterize the surface morphology of thin films and film thickness.

## 3. Results

According to previous studies, when the thickness of the $LaF_3$ film layer increases, the stress of the film layer will gradually increase, and the microstructure defects of the film layer will increase. When the film thickness reaches a certain critical value, the stress even leads to voids, cracks and peeling. Therefore, before exploring the influence of the oxygen content in the film on its infrared optical performance, it is necessary to first explore the appropriate film thickness. As shown in Figure 1a–c, the Ge/$LaF_3$-2(9.60 at.%) is used as the evaporation material, and the monolayer $LaF_3$ infrared film materials with different thicknesses are obtained on the Ge substrate. When the theoretical values of the film thickness are controlled at 300 nm, 500 nm and 700 nm on the vacuum coating equipment, the homogeneous films can be obtained on the Ge substrate. When the cross-section of the film layer is magnified to 20000 times by SEM, the dense films without faults, cracks and other defects can be observed. However, when the thickness is increased to 900nm, it is found that the surface of the Ge substrate is decoated, as shown in Figure S1d. Through SEM analysis of the film side, it can be seen that the average actual thickness is 443 nm, 643 nm and 1040 nm, respectively, as shown in Figure 1d–f.

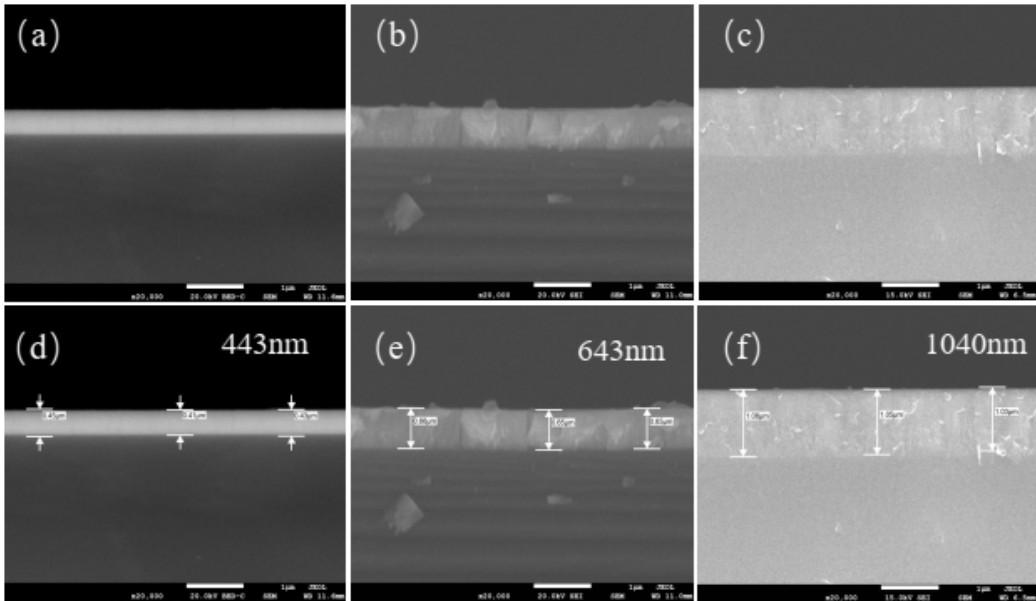

**Figure 1.** Side view of different thickness $LaF_3$ layers. (**a**) and (**d**) 443 nm, (**b**) and (**e**) 643 nm, (**c**) and (**f**) 1040 nm.

Figure 2 shows the transmittance of the $LaF_3$ thin film materials with different thicknesses in the far-infrared region of 8–12 μm, which was obtained by the WQF-510A Fourier transform infrared spectrometer. As can be seen from the Figure, the infrared transmittance of Ge substrate can be improved by coating the $LaF_3$ thin film, the average transmittance increases from 49.9% to 53.4%, and the peak transmittance increases from 50.4% to 57.8%.

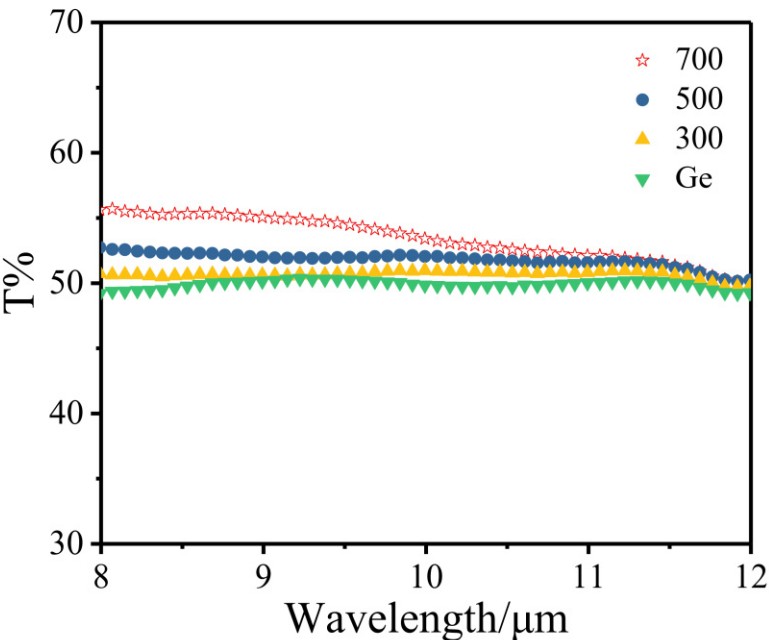

**Figure 2.** The transmittance of Monolayer $LaF_3$ infrared film with different thicknesses (8–12 μm).

To explore the influence of oxygen impurities on the infrared optical properties of the film, the $LaF_3$ crystal materials with different oxygen impurities content are selected as the evaporation source materials. The coating machine sets the theoretical thickness of the film to 700 nm, and then a single-layer $LaF_3$ film material is coated on the cleaned Ge substrate. XRF instrument is used to analyze the surface chemical composition of the film materials [28,29]. Figures S2–S6 show the fluorescence spectra of O elements in the film materials with different oxygen content. According to the peak intensity, with the increase in oxygen content in the raw materials, the peak intensity of O element spectrum is ranked as follows: $Ge/LaF_3$-1 > $Ge/LaF_3$-2 > $Ge/LaF_3$-3 > $Ge/LaF_3$-4 > $Ge/LaF_3$-5, indicating that the content of oxygen impurities in the film decreases accordingly. Figure 3a shows the atomic percentage (at%) of oxygen in the resulting film material, and Figure 3b shows the atomic ratio of O to F. As can be seen from the Figure, the oxygen impurity content in the film material increases from 4.84 at.% to 13.33 at.% with the increase in oxygen impurity content in the raw material. In Figure 3b and Table S2, as the oxygen content of the film decreases, the O/F ratio decreases from 0.160 to 0.055, indicating that the proportion of F increases.

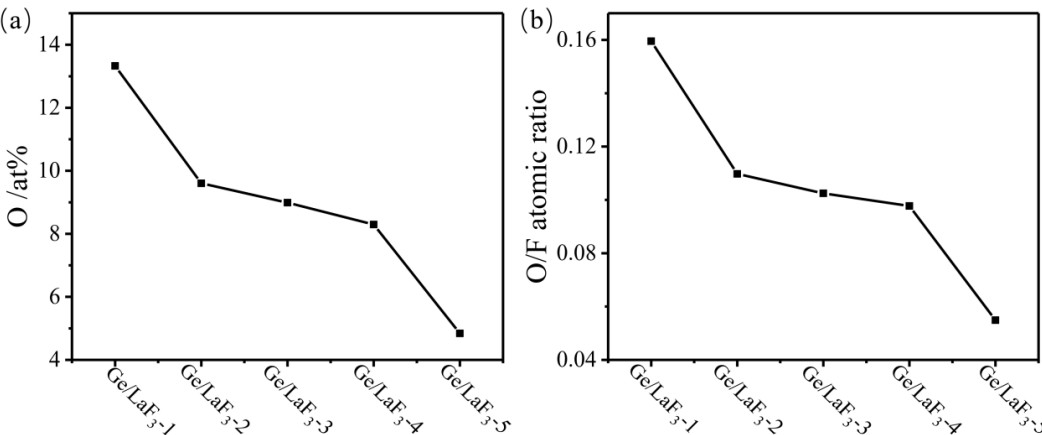

**Figure 3.** Chemical compositions of $LaF_3$ thin films, (**a**) O atomic percentage, (**b**) O/F atomic ratio.

In order to further understand the influence of oxygen impurities on the $LaF_3$ thin film materials, the XRD diffraction patterns of the thin film materials with different oxygen content are studied in this work. Figures S7–S11 show the XRD patterns of Ge/$LaF_3$-1(13.33 at.%), Ge/$LaF_3$-2(9.60 at.%), Ge/$LaF_3$-3(8.99 at.%), Ge/$LaF_3$-4(8.30 at.%) and Ge/$LaF_3$-5(4.84 at.%), respectively. The lattice constants of these three thin film materials are obtained by software fitting, as shown in Table 1, in which $La_2O_3$ (JCPDS No. 05-0602), LaOF (JCPDS No. 44-0121) and $LaF_3$ (JCPDS No. 32-0483) are the standard diffraction cards. The comparison shows that the presence of O impurity destroys the crystal structure of $LaF_3$ film. With the increase in oxygen content, the a-axis and b-axis are compressed to different degrees, while the axis C is extended and the theoretical density decreases and the defects increase.

**Table 1.** The lattice parameters of $LaF_3$ thin film materials.

| Compounds | a | b | c | c/a | Density |
|:---:|:---:|:---:|:---:|:---:|:---:|
| $La_2O_3$ | 3.973 | 3.973 | 6.130 | 1.557 | 6.573 |
| LaOF | 4.088 | 4.088 | 5.811 | 1.421 | 6.010 |
| $LaF_3$ | 7.187 | 7.187 | 7.350 | 1.023 | 5.936 |
| Ge/$LaF_3$-1 | 7.169 | 7.169 | 7.755 | 1.088 | 5.653 |
| Ge/$LaF_3$-2 | 7.173 | 7.173 | 7.524 | 1.049 | 5.822 |
| Ge/$LaF_3$-3 | 7.178 | 7.178 | 7.282 | 1.015 | 6.007 |
| Ge/$LaF_3$-4 | 7.180 | 7.180 | 7.201 | 1.003 | 6.065 |
| Ge/$LaF_3$-5 | 7.183 | 7.183 | 7.107 | 0.990 | 6.146 |

Figure 4 shows the infrared transmittance of the five $LaF_3$ thin films with different oxygen content in the 8–12 μm far-infrared region. Obviously, with the increase in oxygen impurity content, the proportion of F content decreases, and the infrared peak transmittance decreases from 59.9% to 54.5%, while the average transmittance decreases from 58.1% to 52.2%, attributing to the presence of oxygen impurities destabilizing the crystal structure of the $LaF_3$ film materials, resulting in the increase in IR absorption of the film materials. The results are similar to that of others' work [6,30,31], insofar as $CaF_2$ crystal and $YF_3$ thin films are easily contaminated with oxygen. The $YF_3$ molecules combine with oxygen atoms, which leads to the increase in gaps in the film, resulting in the instability of $YF_3$ structure. In order to further clarify the influence of oxygen impurities on the infrared optical properties of the $LaF_3$ film, the refractive index and extinction coefficient of the $LaF_3$ film materials are calculated by the TFCalc software, which can be used to fit the transmittance test data of the film materials, combined with the Cauchy dispersion model and Sellmeier dispersion model; the fitting process is shown in Figure S10. Figure 5 shows the fitting results of the refractive index and extinction coefficient. From the Figure, it can be seen that the refractive index and extinction coefficient of the film layers increase significantly in the 8–12 μm region with the increase in oxygen content in the film materials, and the percentage of F content decreases. In Table S4, the mean refractive index increases from 1.339 to 1.478, and the mean extinction rate increases from 0.001 to 0.030, attributing to the unstable crystal structure of $LaF_3$ due to the presence of O impurities.

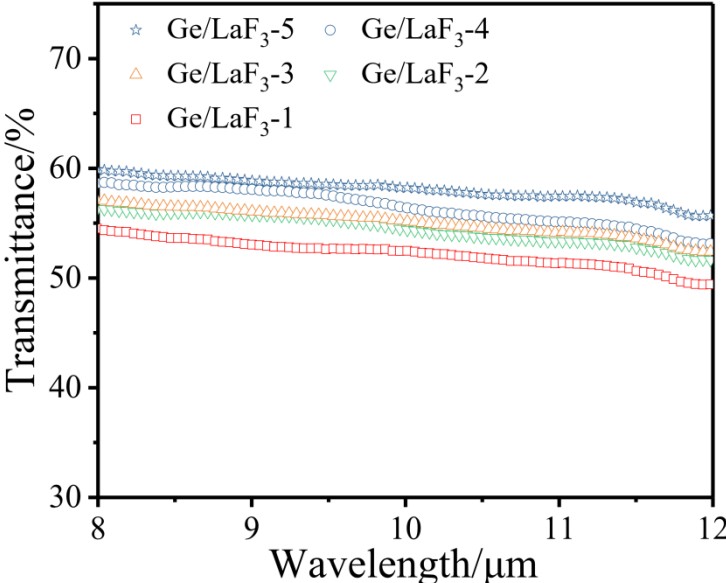

**Figure 4.** The transmission of LaF$_3$ films on Ge substrate with different oxygen contents.

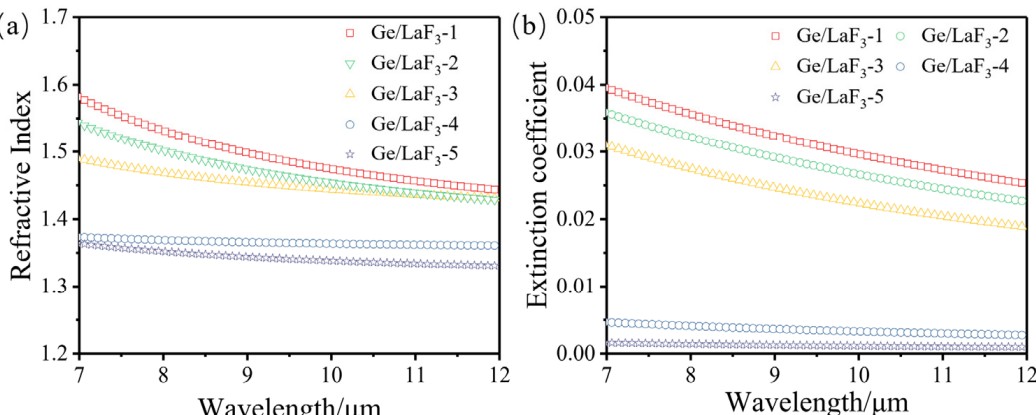

**Figure 5.** (**a**) Refractive index curves and (**b**) Extinction coefficient curves of LaF$_3$ films on Ge substrate with different oxygen contents.

## 4. Conclusions

The influence of oxygen impurity content in the LaF$_3$ infrared antireflection film on its infrared optical properties has been rarely reported. In this paper, the LaF$_3$ crystals with different oxygen contents are used as the coating raw materials, and the LaF$_3$ thin films with different oxygen contents are deposited on the Ge substrates. It is found that with the increase in oxygen impurity content, the proportion of F content decreases, and the crystal structure of the LaF$_3$ film is destroyed, resulting in the decrease in infrared transmittance of the LaF$_3$ film, and the increase in refractive index and extinction coefficient. In order to obtain high-performance of the LaF$_3$ infrared antireflective membrane materials, the content of oxygen impurity in the film material must be controlled strictly and the evaporation process should not be polluted by oxygen. Additionally, the conclusions obtained in this paper lay the theoretical foundation for obtaining high-performance LaF$_3$ film materials.

**Supplementary Materials:** The following supporting information can be downloaded at: https://www.mdpi.com/article/10.3390/coatings12081184/s1, Figure S1: Apparent morphologies of LaF3 thin films with different thicknesses, (a) 300 nm, (b) 500 nm, (c) 700 nm and (d) 900 nm; Figure S2: XRF spectra of O in Ge/ LaF3-1 thin films, O/F ratio is 0.160; Figure S3: XRF spectra of O in Ge/ LaF3-2 thin films, O/F ratio is 0.110; Figure S4: XRF spectra of O in Ge/ LaF3-3 thin films, O/F ratio is 0.102; Figure S5: XRF spectra of O in Ge/ LaF3-4 thin films, O/F ratio is 0.098; Figure S6: XRF spectra of O

in Ge/ LaF3-5 thin films, O/F ratio is 0.055; Figure S7: XRD patterns of the Ge/LaF3-1(13.33 at.%) thin film; Figure S8: XRD patterns of the Ge/LaF3-3(9.60 at.%) thin film; Figure S9: XRD patterns of the Ge/LaF3-5(8.99 at.%) thin film; Figure S10: XRD patterns of the Ge/LaF3-4(8.30 at.%) thin film; Figure S11: XRD patterns of the Ge/LaF3-5(4.84 at.%) thin film; Table S1: LaF3 crystal raw materials with different oxygen content; Table S2: The O atomic percentage and O/F atomic ratio of LaF3 thin films; Table S3: The peak transmission and average transmittance of LaF3 films on Ge substrate with different oxygen contents; Table S4: The refractive index and extinction coefficient of $LaF_3$ films on Ge substrate with different oxygen contents.

**Author Contributions:** Conceptualization, W.L. and C.Y.; Methodology, W.L., C.Y., D.C. and Z.W.; Software, W.L.; Formal analysis and investigation, B.P. and D.Z., L.H.; Writing—original draft preparation, W.L.; Writing—review and editing, W.L., R.M. and B.Y.; Funding acquisition, R.M. All authors have read and agreed to the published version of the manuscript.

**Funding:** This study is financially supported by the Key R & D Plan of Hebei Province (19211503D) and Aero Engine and Gas Turbine Major Project (J2019-VI-0023-0140). Engineer W.L. Lu is supported by Youth Talent Lifts plan of Grinm Group Co., Ltd. (0111970400364).

**Institutional Review Board Statement:** Not applicable.

**Informed Consent Statement:** Not applicable.

**Data Availability Statement:** Not applicable.

**Conflicts of Interest:** The authors declare no conflict of interest.

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
