# Peer review of "Dependence of Structural and Optical Performance of Lanthanum Fluoride Antireflective Films on O Impurities"

_coatings, doi:10.3390/coatings12081184_

Round 1

Reviewer 1 Report

This version of the manuscript does not look worthy and cannot be recommended for publication in this form and at least needs some revision.

1.     Line 34-37.   Additional introductory information about LaF3 is needed not only to better reveal the motivation of the work, but also to attract a wider readership. See for example,

see the following paper and references therein:

Chuklina, N.; Piskunov, S.; Popov, N.; Mysovsky, A.; Popov, A. Comparative quantum chemistry study of the F-center in lanthanum trifluoride. Nucl. Instrum. Methods Phys. Res. B 2020474, 57–62.

LaF3 is also one of the main components of the newly developed fluoride-glass ceramic.

E. Elsts, et al , "Fast Luminescence Studies of NaLaF4: Pr3+ Glass Ceramics," 2021 IEEE 12th International Conference on Electronics and Information Technologies (ELIT), 2021, pp. 287-290,

https://doi.org/10.1109/ELIT53502.2021.9501063\

2.     Line 42-45. These sentences require additional analysis, because they contain incomprehensible grammatical phrases.

3.     Analysis of the list of references for the introduction shows that the motivation and relevance of the work are not sufficiently disclosed. Most of the supporting references are quite old.

4.     Speaking of oxygen centers in LaF3, it will be very important to see a more detailed analysis of their structure. See as an example: https://doi.org/10.1103/PhysRevB.84.064133 and references therein.

5.     Detailed short comparative analysis of oxygen centers in other fluorides is essential !!!

6.     Why can't you see OH in the spectra?

Author Response

Comment 1: Line 34-37. Additional introductory information about LaF3 is needed not only to better reveal the motivation of the work, but also to attract a wider readership. See for example, see the following paper and references therein:

Chuklina, N.; Piskunov, S.; Popov, N.; Mysovsky, A.; Popov, A. Comparative quantum chemistry study of the F-center in lanthanum trifluoride. Nucl. Instrum. Methods Phys. Res. B 2020, 474, 57–62.

LaF3 is also one of the main components of the newly developed fluoride-glass ceramic. Elsts, et al , "Fast Luminescence Studies of NaLaF4: Pr3+ Glass Ceramics," 2021 IEEE 12th International Conference on Electronics and Information Technologies (ELIT), 2021, pp. 287-290, https://doi.org/10.1109/ELIT53502.2021.9501063\.

Response: Thanks a lot for your helpful suggestion. Lanthanum fluoride has a wide range of applications which are supplemented in this paper.

Please see: Main text, the section 1, the first paragraph, line 35-39.

Comment 2: Line 42-45. These sentences require additional analysis, because they contain incomprehensible grammatical phrases.

Response: Thanks for your comments. These sentences have been revised carefully.

Please see: Main text, the section 1, the second paragraph, line 42-59.

Comment 3: Analysis of the list of references for the introduction shows that the motivation and relevance of the work are not sufficiently disclosed. Most of the supporting references are quite old.

Response: Thank you for your pertinent advice. In the article, we supplement some relevant papers which were published in the last three years. Moreover, in the second paragraph of section 1, we introduce the relevant work of lanthanum fluoride infrared antireflective coating materials in chronological order. Therefore, the older literatures were selected. Finally, in the field of infrared antireflective coating materials, researchers pay more attention to YbF3 and YF3. LaF3 is a novel coating material, so there are few references.

Please see: Ref. 18, Ref. 20-23.

Comment 4: Speaking of oxygen centers in LaF3, it will be very important to see a more detailed analysis of their structure. See as an example: https://doi.org/10.1103/PhysRevB.84.064133 and references therein.

Response: Thanks for reminding me. In this reference, the authors used an embedded cluster and ab initio calculations. These methods are useful, but too specialized. And We do not have the corresponding conditions to use the same methods to analyze LaF3 films in this article. But this reference shows the reliability of our conclusion from the side. Oxygen impurities will affect the structure of LaF3 film, and then affect the optical properties of the film. We will cite it in our article.

Please see:Ref. 29.

Comment 5: Detailed short comparative analysis of oxygen centers in other fluorides is essential !!!

Response: Thanks for reminding me. It is shown in Table 1 that the effect of oxygen impurities on the crystal results of other fluoride films. Figure S8 and S10 show the XRD patterns of the Ge/LaF3-2(9.60at%) and Ge/LaF3-4(8.30at%) thin films respectively. At the same time, as a footnote, we show the comparative analysis of oxygen centers in YF3 film in line 167-169.

Comment 6: Why can't you see OH in the spectra?

Response: Thanks for reminding me. Infrared transmission for the films deposited on Ge was measured from 8μm to 12μm. Incorporation of molecular level of O-H bonds is apparent by absorption regions at 2.8-3.2μm (stretching absorption band) and 5.6-6.5μm (bending absorption band). Please see Ref. 22.

Reviewer 2 Report

The paper is interesting from the material point of view. I miss an improved description explaining the aim and the application field of this work.

In addition, I have several comments:

- Refering to the tendency observed in SEM images, I miss the same analysis with the samples with the highest and lowest oxygen concentration. Is the tendency about the cracks, faults and other defects the same if the oxygen concentration?

-Where is the peak of transmittance? I don't find it. Is this tendency regardless of the oxygen concentration?

-How is possible to measure oxygen concentration in a quantitative way with XRF? This is the more important part of the paper, and from my point of view, it is difficult to obtain a numeric value of the oxygen concentration with this technique; is there any possibility to check these values?

-How do you obtain the density? could you explain it in the text?

-Fig 4: thickness? is the same for all the samples?

Author Response

Reviewer 2: The paper is interesting from the material point of view. I miss an improved description explaining the aim and the application field of this work. In addition, I have several comments:

Comment 1: Refering to the tendency observed in SEM images, I miss the same analysis with the samples with the highest and lowest oxygen concentration. Is the tendency about the cracks, faults and other defects the same if the oxygen concentration?

Response: Thank you for your comment. In LaF3 film, the cracks, faults and other defects are mainly caused by the internal stress and have little relation with the oxygen concentration. The same phenomenon can be observed in Cheng’s work (Ref. 14).

Comment 2: Where is the peak of transmittance? I don't find it. Is this tendency regardless of the oxygen concentration?

Response: Thank you for your comment. In line 163-164, the peak transmittance represents the maximum transmittance of the film materials in 8-12μm and is not the peak of the transmittance spectra. In Table S3, the peak transmittances of LaF3 film materials with different oxygen content are shown. With the decrease of O content, the peak transmittance increases from 54.5% to 59.9%. Please see this analysis in line 162-165.

Comment 3: How is possible to measure oxygen concentration in a quantitative way with XRF? This is the more important part of the paper, and from my point of view, it is difficult to obtain a numeric value of the oxygen concentration with this technique; is there any possibility to check these values?

Response: Thank you for your kind suggestion. It has been reported that XRF method was used to detect the oxygen content in relevant literature and standards. In the book of Modern Optical Thin Film Technology(p435), the authors stated that the spectral levels and intensities emitted by various photons (X-rays), electrons and ions bombarding the film can be used to determine the content of various components of the film. (Please refer to QING JINSHU[J]. 2021, 2, 51-55. Proceedings of the 8th National Symposium on X-ray Fluorescence Spectroscopy[C]. 2010, 76. Italian Industry Standard, UNI EN 725-4-2006.)

Comment 4: How do you obtain the density? could you explain it in the text?

Response: Thank you for your comment. In line 153-154, when the JADE software was used to fit the diffraction peaks of the film, we can not only get the lattice constants a, b and c, but also obtain the theoretical density of the thin films, as shown in the Figure below. Please see the book named as “X-ray diffraction of polycrystalline materials - Experimental principles, methods and applications” (Jiwu Huang, Metallurgical Industry Press)

Comment 5: Fig 4: thickness? is the same for all the samples?

Response: Thank you for your comment. In the process of film preparation, except for the difference of evaporation source materials, the other technological parameters remain the same, so the thickness of the film obtained is basically not different. In the below figure, the coating machine sets the theoretical thickness of the film to 700nm, the Ge/LaF3-1(13.33at%), Ge/LaF3-3(8.99at%), Ge/LaF3-4(8.30at%) and (d) Ge/LaF3-5(4.84at%) were obtained. The thickness of the layer was measured by SEM. It can be obviously found that the thickness of different LaF3 layers is about the same.

Reviewer 3 Report

The paper is based on the structural and optical properties dependance on O impurities in lanthanum fluoride antireflecting films and characterized with the techniques like XRD, SEM, and FTIR study. The study is not complete with lack of analysis on the influence of Oxygen as compared to other materials. The authors should pay attention to the in-depth discussion of the optical properties change and also provide the XRD data which is missing. Since the title includes structural properties and without XRD discussion, the study is incomplete. Overall, the manuscript is poorly written and needs systematic analysis and discussion of the results in light of the title. So, my recommendation is to Reject the manuscript.

Author Response

Reviewer 3: The paper is based on the structural and optical properties dependance on O impurities in lanthanum fluoride antireflecting films and characterized with the techniques like XRD, SEM, and FTIR study. The study is not complete with lack of analysis on the influence of Oxygen as compared to other materials. The authors should pay attention to the in-depth discussion of the optical properties change and also provide the XRD data which is missing. Since the title includes structural properties and without XRD discussion, the study is incomplete. Overall, the manuscript is poorly written and needs systematic analysis and discussion of the results in light of the title. So, my recommendation is to Reject the manuscript.

Response: Thank you for your kind suggestions. It is shown in Table 1 that the effect of oxygen impurities on the crystal results of other fluoride films. Figure S8 and S10 complementally show the XRD patterns of the Ge/LaF3-2(9.60at%) and Ge/LaF3-4(8.30at%) thin films respectively. At the same time, as a footnote, we show the comparative analysis of oxygen centers in CaF2 crystal and YF3 film in line 167-169. The structure in the title represents the crystal structure of the LaF3 films. In line 152-159, Table 1 and Figure S7-S11, the changes of the films’ crystal structure were fully shown with the increase of oxygen content.

Round 2

Reviewer 1 Report

the authors have successfully responded to all the recommendations of the reviewer, so the article can be recommended for publication as it is.

Reviewer 3 Report

The authors have revised accordingly and improved the manuscript.